# Sexual Dimorphism in the Multielemental Stoichiometric Phenotypes and Stoichiometric Niches of Spiders

**DOI:** 10.3390/insects11080484

**Published:** 2020-07-30

**Authors:** Łukasz Sobczyk, Michał Filipiak, Marcin Czarnoleski

**Affiliations:** Institute of Environmental Sciences, Jagiellonian University, ul. Gronostajowa 7, 30-387 Kraków, Poland; lukasz.sobczyk@uj.edu.pl (Ł.S.); marcin.czarnoleski@uj.edu.pl (M.C.)

**Keywords:** ecological stoichiometry, predator, spider, sex, nutrition, nutritional ecology, arthropod, nutrient cycling, trophic link, food web

## Abstract

Nutritional limitations may shape populations and communities of organisms. This phenomenon is often studied by treating populations and communities as pools of homogenous individuals with average nutritional optima and experiencing average constraints and trade-offs that influence their fitness in a standardized way. However, populations and communities consist of individuals belonging to different sexes, each with specific nutritional demands and limitations. Taking this into account, we used the ecological stoichiometry framework to study sexual differences in the stoichiometric phenotypes, reflecting stoichiometric niches, of four spider taxa differing in the hunting mode. The species and sexes differed fundamentally in their elemental phenotypes, including elements beyond those most commonly studied (C, N and P). Both species and sexes were distinguished by the *C*:*N* ratio and concentrations of Cu, K and Zn. Species additionally differed in concentrations of Na, Mg and Mn. Phosphorous was not involved in this differentiation. Sexual dimorphism in spiders’ elemental phenotypes, related to differences in their stoichiometric niches, suggests different nutritional optima and differences in nutritional limitation experienced by different sexes and species. This may influence the structure and functioning of spider populations and communities.

## 1. Introduction

### 1.1. Framework of Ecological Stoichiometry May Be Applied for a Better Understanding of the Ecology and Evolution of Organisms

Energy supply and demand are widely considered to affect the evolution of life histories [1], and to date, many studies have focused on the effects of adaptations on energy balance, the efficiency of energy acquisition and investment, and limits to energy budgets [1,2,3]. Equally important but less studied is the need to maintain stoichiometric balance [4,5,6]. Among heterotrophs, the mismatch between the chemical compositions of consumer tissues and consumer food can strongly affect major life history traits (e.g., growth rates, body size, reproductive strategies and survival) [7]. Therefore, the energy-oriented view of the diversity of life is incomplete, as it considers energy budgets as the sole factor limiting the capacity to produce new tissue; luckily, ecological stoichiometry allows for considering both energy and matter in studies related to the ecology and evolution of organisms [4,7,8,9].

Ecological stoichiometry considers the growth and development of every cell, tissue, organism and population to be subject to the law of conservation of mass [7], which certainly requires myriads of biochemical reactions to be chemically balanced. Within this framework, the concepts of the biogeochemical niche and stoichiometric niche were recently proposed [10,11]. Both concepts similarly acknowledge that the availability of particular atoms in specific proportions is a selective factor that affects the evolution of life and the environment (hereafter, the term “stoichiometric niche” will be used). Organismal stoichiometry, also called the elemental phenotype or elementome, determines the organism’s demand for resources used for growth and development [11,12,13] and is highly dependent on the trophic position and phylogeny [14,15,16]. Following this, the stoichiometric niche is defined as a multivariate niche space occupied by a group of individuals with similar stoichiometric phenotypes, with specific species occupying specific niches [10,11]. To obtain stoichiometrically balanced food, various organisms with specific stoichiometric phenotypes might prefer various food sources that provide nutrients in proportions that reflect their nutritional demand [7,8,13]. It should be stressed here that a stoichiometric mismatch between the elemental phenotype and food should be primarily considered with reference to the production of new tissue, either the tissue of the organism during growth and development life stages or the tissue of offspring in reproducing organisms. Additionally, more precise evaluations of whether the stoichiometry of food is optimally balanced should not only consider the current chemical composition of the body but also the need to utilize a part of the acquired carbon for ATP production and turnover rates of elements in a tissue. Demand for carbon that covers energetic needs is relatively well studied; for example, research on arthropods suggests that approximately 75% of carbon in food is respired and released from the body in the form of CO_2_ [8]. Although obtaining information on the turnover rates of elements is difficult, certain elements should have higher turnover rates compared to other elements and these rates are likely to differ among species [17].

Stoichiometric mismatch between the atomic ratios of an organism’s body and its food is expected to have negative fitness consequences [4,13,18,19,20]. Considering the framework of ecological stoichiometry, the quantity of food provided to an organism cannot be a substitute for its quality, and the observed toxic effects of a particular diet on an organism may be caused by stoichiometric mismatch rather than by toxic substances [5,21]. The limiting effects of a stoichiometrically unbalanced diet may be scaled-up from individuals to entire communities [7]. In this way, populations and communities of organisms may be shaped by nutritional limitations resulting from stoichiometric mismatches.

### 1.2. Sexual Dimorphism in Organismal Stoichiometry Is an Underrated But Important Component of the Functioning of Populations and Communities of Organisms

Differences in organismal stoichiometry are usually studied by comparing species, but researchers have started to pay attention to the origin of individual variation in the chemical composition of a body, showing that intraspecific differences in organismal stoichiometry may originate from sexual dimorphism, among other sources [22,23,24]. Such differences are associated with the differential effects on females and males of the processes involved in life history evolution and population dynamics because these effects impose sex-specific nutritional limitations [13,23,24]. Therefore, the categorical statement that a species is limited by the nutritional quality of available resources may be disconnected from natural situations, and the results of ecological and evolutionary modeling may be biased if interspecific variability is not considered [23,25]. Recently, research has begun to focus on individual variation in the chemical composition of the body, considering that processes involved in life history evolution and population dynamics are likely to differentially affect females and males, imposing sex-specific nutritional limitations; however, such studies are limited in number [13,22,23,24]. Importantly, by considering such within-species variance, evaluations of resource limitations in a given species increase in ecological relevance.

### 1.3. Goal of the Study

Recent studies have shown that a better understanding of nutritional relations between trophic links in terrestrial communities is needed to explain the observed patterns of trophic interactions. Within this context, it was pointed out: (1) that our knowledge of the nutritional ecology of trophic interactions is biased towards herbivores and omnivores [26], (2) that the need for nutrient regulation in carnivores is routinely ignored because predator food is assumed to be of an overall high quality and low nutritional variability [27], (3) that new knowledge on stoichiometric components in predator–prey interactions is needed to connect basic physiological mechanisms underlying predator nutritional needs with prey physiological responses to predation risk and (4) that a deeper understanding of nutritional interactions among species, communities and guilds is needed [26,28]. After 30 years of development of the framework of ecological stoichiometry and almost 20 years after the seminal book of Sterner and Elser [7] was published, data on the elemental phenotypes of a variety of organisms are scarce and lag far behind the data on their genomes [29]. Moreover, arachnids, including spiders, are particularly suited for studies of the general patterns in sexual dimorphism [30,31]. Our work attempts to fill some of the gaps in knowledge mentioned above by considering the stoichiometric relationships between 11 elements composing the bodies of two sexes in different spider taxa, representing different hunting strategies. We studied body stoichiometry in different groups of spiders to address for the first time taxon-specific and sex-specific dimorphism in the multielemental stoichiometric phenotype and stoichiometric niche of spiders with different hunting modes.

### 1.4. Hypotheses

We measured the concentrations of 11 elements (C, N, P, K, Na, Ca, Mg, Fe, Zn, Mn and Cu) in the bodies of both sexes of six spider taxa, each of which had a different hunting strategy (among which four taxa could be used in the following statistical analysis). Processes involved in ecophysiology and life history evolution are likely to differentially affect females and males, imposing sex-specific nutritional limitations and even imposing a conflict between the sexes in reaching sex-specific nutritional optima [13,23]. At the same time, stoichiometric phenotypes are species-specific and reflect the nutritional demands of individuals [15,24,32]. Therefore, we hypothesized (1) that taxon-specific and sex-specific differences in the multielemental stoichiometry of the studied spiders and in their stoichiometric niches would be observed. It was previously suggested, considering various organisms beyond spiders, that females have a higher demand for C and P than males since the former produce more P-rich nucleic acids and have a greater need for lipids [7,13,33]. Therefore, we hypothesized (2) that sexual dimorphism would be mainly related to larger proportions of C and P in the bodies of females than in those of males. Due to the scarcity of related data, it was impossible to hypothesize specific differences related to the body stoichiometry of different spider taxa.

## 2. Materials and Methods

Spiders were collected within the city limits of Kraków, Poland (50°03′41″N, 19°56′11″E; elevation: 219 m.a.s.l.). They represented six hunting models: (1) orb-weaver spiders (Araneidae, genus *Araneus*), represented by 85 females and 14 males; (2) jumping spiders (Salticidae), represented by 193 individuals (54 males, 44 females and 95 spiders of unrecognized sex); (3) crab spiders (Thomisidae), represented by 326 individuals (unrecognized sex), mainly from the *Misumena* genus; (4) wolf spiders (Lycosidae), represented by 56 individuals (unrecognized sex); (5) funnel weavers (Agelenidae, genus *Tegenaria*), represented by 31 females and 15 males and (6) cobweb spiders (Theridiidae, *Steatoda grossa*), represented by 103 females and 149 males. All the collected individuals were used to prepare analytical samples, for which the measured values of elemental concentrations are presented in Appendix A and Table A1. Ultimately, for the purpose of this study, we analyzed data only for spiders for which we were able to obtain sex-specific analytical measurements, namely, (1) orb-weaver spiders in *Araneus*, (2) jumping spiders in Salticidae, (5) funnel weavers in *Tegenaria* and (6) cobweb spiders of *Steatoda grossa*. Raw data on stoichiometry for the two remaining groups (3, 4), for which sex-specific information was not available, were only included in Table A1 for reference. It is important to mention that the reproductive state of collected females was not studied here; thus, females with or without eggs in their bodies could contribute to our samples, which could have increased the variation in the measured concentrations of elements.

Considering the detection limits of the analytical equipment, we measured the concentrations of the studied elements in samples comprising of one to one hundred individuals, depending on their body mass. The number of individuals comprising every analytical sample is presented in Table A1.

To create analytical samples, spiders were grouped according to sex and species (if identification was possible) or higher taxonomic level (if information on species identity was unavailable). Each sample was freeze-dried, ground and homogenized by a mortar and then freeze-dried again to obtain dry mass for elemental analyses. From each dry mass sample, two analytical subsamples were obtained: (i) a liquid solution subsample used to analyze K, Ca, Mg, Fe, Zn, Mn, Cu, Na and P (hotplate acid digestion with a 4:1 solution of nitric acid (70%) and hydrogen peroxide (30%)) and (ii) a dry mass subsample used to analyze C and N. The concentrations of C and N were determined with a Vario EL III automatic CHNS analyzer; the concentrations of K, Ca, Mg, Fe, Zn, Mn, Cu and Na were determined by atomic absorption spectrometry (Perkin-Elmer AAnalyst 200 and AAnalyst 800); and the concentration of P was determined by colorimetry (MLE FIA flow injection analyzer). Sulfanilic acid was used as the reference material for the C, and N analyses, and certified reference materials (NCSDC73349, NCSZC73016 and RM8415) were used for the other elements.

Principal component analysis (PCA) was employed to reconstruct the multielemental stoichiometric relations among taxa and sexes using Canoco 5 [34]. The data were log-transformed, centered and standardized by species but not by sample; thus, PCA was performed on a correlation matrix. To check for differences between sexes and taxa, we performed analysis of variance (ANOVA) independently for the 1st- and 2nd-axis scores using Statistica 13 (TIBCO Software Inc.).

## 3. Results

On the plane determined by the first two axes (together explaining 63.19% of the total variance), spiders largely grouped according to taxon identity and sex (Figure 1). The 1st axis was loaded mostly by the variance in N (loading value: 0.93), Cu (0.91), K (0.84), C (0.82) and Zn (0.76; see Table 1 for more loading values). Following our ANOVA of the 1st axis scores (Table 2), these five elements corresponded the most to the observed differentiation between sexes, with females having higher concentrations of C and lower concentrations of N, Cu, K and Zn than males (Figure 1 and Figure 2). The observed pattern was driven principally by the *C*:*N* ratio, as shown by the parallel C and N vectors pointing in opposite directions (Figure 1). Therefore, females had significantly higher *C*:*N* ratios but not *C*:*P* and *N*:*P* ratios than males. Similarly, the concentrations of these five elements differed between *Steatoda grossa* (higher C concentration and lower concentrations of N, Cu, K and Zn) and other taxa of the studied spiders (lower C concentration and higher concentrations of N, Cu, K and Zn; Figure 1 and Figure 3). The 2nd axis was loaded primarily by the variance in Na (loading value: 0.95), Mn (0.63) and Mg (0.54; see Table 1 for more loading values). Following our ANOVA of the scores of the 2nd axis (Table 2), these three elements corresponded the most to the observed differentiation between taxa, with *Steatoda grossa* and *Tegenaria* having higher body concentrations of Na and Mg but lower concentrations of Mn than *Araneus* and Salticidae (Figure 1 and Figure 4).

## 4. Discussion

Our results supported hypothesis (1), i.e., the view that stoichiometric phenotypes differ not only among taxonomic groups but also between sexes. This suggests that different taxa as well as different sexes tend to occupy distinct multielemental stoichiometric niches. Importantly, the nature of stoichiometric differences between sexes was similar among spider taxa, irrespective of their hunting modes. In particular, females were characterized by a higher concentration of C and lower concentrations of N, Cu, K and Zn than males. This pattern was shaped mostly by the C:N ratio, which was higher for females than for males. Since the P concentration was similar in the bodies of both sexes, we found only partial support for hypothesis (2); i.e., females had a higher concentration of C in their bodies than males, but the P concentration was similar between the sexes; additionally (which was not hypothesized), females had lower concentrations of N, Cu, K and Zn than males. Importantly, we have shown that multiple elements beyond C, N and P shape stoichiometric niches.

To date, it has been shown that spider species specializing in different web architectures differ in body stoichiometry regarding C:N and N:P ratios but not the C:P ratio and in the concentration of N but not C or P [35]. Our results conform to these findings by showing stoichiometric differences among spiders having different hunting strategies, including those building webs and hunting without webs. In our study, similar to the findings of Ludwig and colleagues [35], spider species differed mainly due to different C:N ratios; however, we show that the stoichiometric niche of the species was more complex and additionally structured by concentrations of Cu, K, Zn, Na, Mg and Mn. Interestingly, we found that *S. grossa* females had exceptionally high C:N and C:Zn ratios compared with other studied spiders, and increased in the C:other elements ratio could reflect rich fat deposits in a body. However, increased fat deposition should result in a proportional increase of the ratio of C to all other elements, which is not reflected in our results. The observed pattern could reflect additional factors, including a specific reproductive stage of *S. grossa* individuals, e.g., the investment in egg production. Undoubtedly, future studies should more directly address how reproductive states affect the elemental content of the body. What is exceptionally interesting is that sex differences are similarly driven mainly by differences in the *C*:*N* ratio (higher in females), followed by differences in the concentrations of Cu, K and Zn. Additionally, the sexual differences in spiders’ stoichiometric phenotypes differed from multielemental phenotypes reported for other arthropods, i.e., two bee species [24,36] and three species of beetles [37]. Therefore, a multielemental picture of stoichiometric differences between sexes may be taxonomically specific and/or feeding-guild-specific rather than uniform among taxonomical groups and/or feeding guilds. We conclude that in studies dealing with ecological stoichiometry, samples should be divided according to sex because under/overrepresentation of a specific sex may influence the results. Importantly, the observed patterns suggest that stoichiometric relations between atoms of more elements than the commonly studied C, N and P may shape ecological interactions and the functioning of food webs, which is also confirmed by the results of the limited studies on multielemental ecological stoichiometry published to date (e.g., [8,24,38]). Recently, the role of Na limitation in shaping ecological interactions was emphasized [39]; however, approximately twenty-five elements build molecules driving the functioning of every living organism on Earth, and much research is still needed in the area of multielemental stoichiometry [5,8,29]. This is made even more complicated by the fact that the concentration of every element composing the organismal body influences the concentrations of all other elements, and all of them are strongly interrelated [38]. Therefore, it is essential to consider multielemental ratios in studies dealing with ecological stoichiometry.

Only a few studies to date have considered the ecological stoichiometry and elemental nutrition of spiders (e.g., [32,35,40,41,42]); therefore, a large gap in data exists, and it is hard to perform comparisons leading to strong and meaningful conclusions. Hence, we will supplement our reasoning with available data collected for other arthropods. A study on different “populations” (natural and reared in the laboratory) of the black widow spider *Latrodectus hesperus* suggested between-“population” differences in C:N, C:P and N:P stoichiometries [40]. However, only female spiders were considered. Our current study does not allow us to compare variation in stoichiometric phenotypes between sexes because of the too small number of analytical samples collected for males; however, it was previously shown, using solitary bees as model organisms, that females are more homeostatic than males in their elemental phenotype [24]. If so, it may be possible to find statistically significant differences in stoichiometric phenotypes of females but not males associated with different populations. Considering that sexual dimorphism may apply to both the specific stoichiometric phenotype and the level of homeostasis of this phenotype, we conclude that the results obtained for only one sex should not be approximated to the whole population.

Different sexes are exposed to distinct selection pressures posing sex-specific constraints on individuals and have different nutritional optima, and they may even experience conflicting selection pressures [43,44]. Moreover, food quality may drive sexually different selection pressures to a greater extent than food quantity, since definite nutrients in exact proportions are needed to build particular body structures [13,23,45]. Correspondingly, we observed fundamental differences in the elemental phenotype between male and female spiders, which might suggest sex-specific challenges in the environment and thus selection pressures. This observation means that the two sexes of spiders might differ in a stoichiometric niche that reflects sex-specific nutritional demands. Therefore, different proportions of nutrients in food might be optimal for females and males. This finding has important implications for studying the ecology and evolution of sexual dimorphism. Considering nutrition, many animals are known to sexually differ in their trophic behavior (e.g., foraging for different foods), morphology (e.g., specialized morphological structures) or physiology (e.g., males do not eat at all), which are all expressed as easily observable traits [46]. Our study shows that these differences may be ecologically and evolutionarily relevant even if they are imperceptible, similar to the example of differences in stoichiometric niches. Stoichiometric niches induce resource competition, resulting in nutritional limitation that in turn shapes populations and communities of organisms [7]. However, most stoichiometrically explicit models do not consider population and community structure, where intraspecific variability in elemental stoichiometry exists [43,47,48,49]. Concurrently, organismal fitness may be influenced by trade-offs originating from the conflict between the sexes in reaching sex-specific nutritional optima [22,23,50]. Hence, the observed sexual dimorphism in stoichiometric niches may shape spider population structure via the nutritional quality of available food.

Our results also have implications for the conservation biology of spiders. Recently, it was suggested that spiders are threatened by human-induced erosion of trophic webs [51]. Nyffeler and Bonte [51] found a dramatic population density decline for the spider *Araneus diadematus* in Swiss Midlands and discussed the ongoing abundance decline of spiders and other insectivorous animals. They related the decline in spider abundance to the decline in insect abundance. Our study suggests a more complicated picture, where the direct driver of spider decline is not the total insect decline but declines in the proportions of specific spider food sources that are stoichiometrically balanced for spiders and part of sex-specific and species-specific stoichiometric niches.

## 5. Conclusions

Populations are not homogenous but consist of individuals belonging to different sexes, which have distinct stoichiometric niches. Intraspecific variability in elemental stoichiometry has rarely been studied, and most stoichiometrically explicit models treat populations/communities as pools of uniform elements, ignoring natural population/community structure. Therefore, we predict that underestimating variability among individuals in populations might influence conclusions about the outcomes of resource limitation in nature. At the same time, the conflict between sexes in reaching sex-specific nutritional optima or stoichiometric constraints posed on a single sex may be crucial for population growth. We believe that considering such within-species variance will make evaluations of resource limitations in a given species more ecologically relevant. Moreover, multielemental stoichiometry beyond *C*:*N*:*P* should be considered to fully understand the complex relationships of the ratios of all 25 (approximately) atoms building the physiological machinery and morphological traits of every organism on the planet.

## Figures and Tables

**Figure 1 insects-11-00484-f001:**
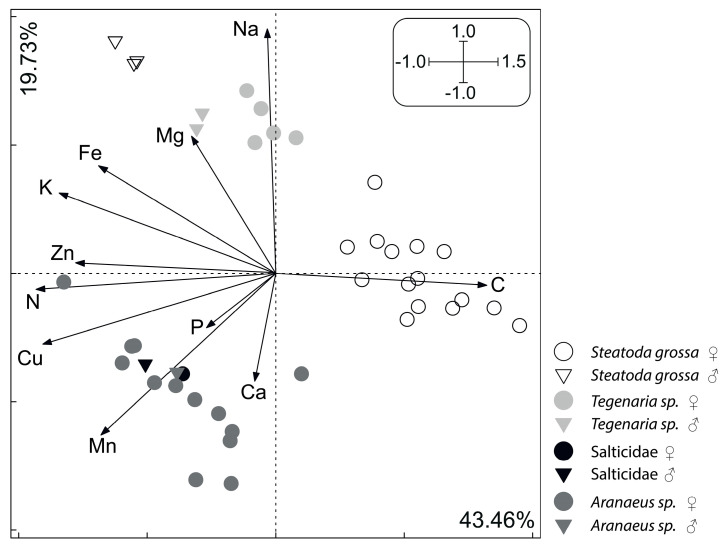
Principal component analysis (PCA) plot—multivariate analysis of stoichiometric relations in spider taxa and sexes. The first two axes are presented. Considering the 1st axis, females are separated from males primarily due to their relatively high concentration of C and low concentrations of N, Cu, K and Zn (see Table 2 and Figure 2 for detailed statistics). Similarly, *Steatoda grossa* is separated from the other taxa due to its relatively high concentration of C and low concentrations of N, Cu, K and Zn (the other taxa do not differ statistically from each other; see Table 2 and Figure 3 for statistics). Considering the 2nd axis, *Tegenaria* sp. forms a separate cluster mainly due to its relatively high concentrations of Na and Mg and low concentration of Mn. An opposite tendency is observed for *Araneus* sp., while *Steatoda grossa* together with Salticidae forms one cluster with intermediate concentrations of Na, Mg and Mn. These tendencies were confirmed by ANOVA performed independently for the 1st and 2nd axis scores (*p* < 0.05; Figure 2, Figure 3 and Figure 4).

**Figure 2 insects-11-00484-f002:**
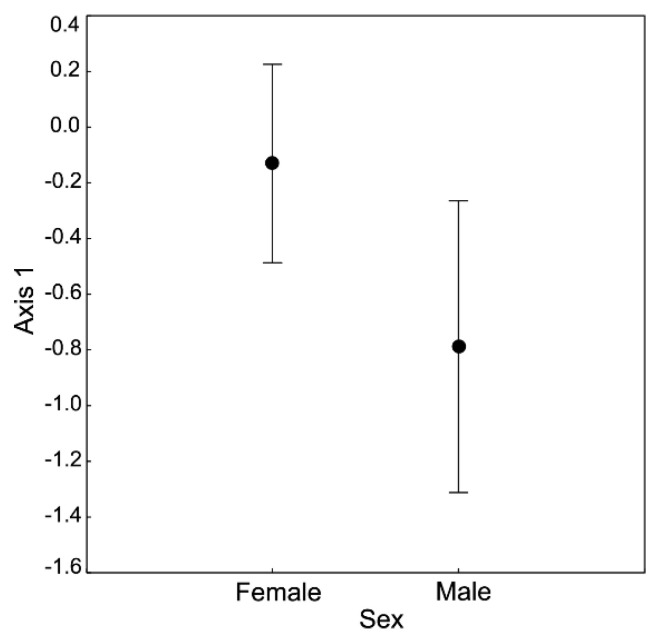
Multivariate analysis of stoichiometric relations in two sexes of spiders (PCA). ANOVA performed for the 1st axis scores (F = 4.46, *p* = 0.04). Bars denote the confidence intervals. Table 1 provides information on the contribution of different elements to the scores of Axis 1.

**Figure 3 insects-11-00484-f003:**
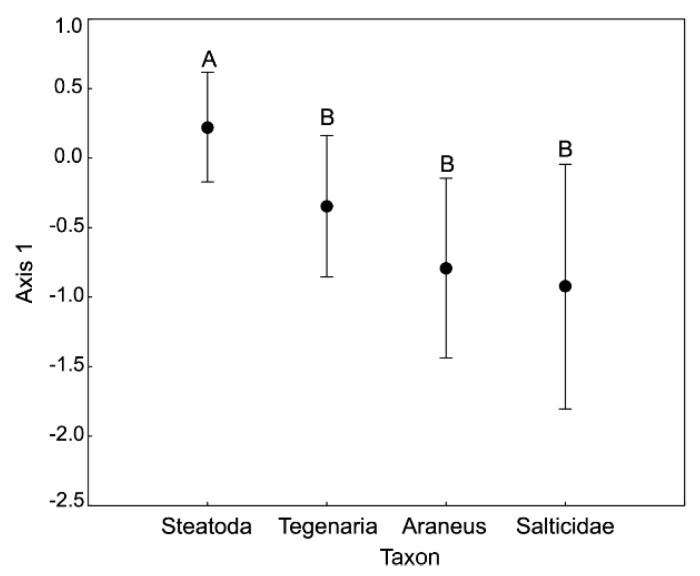
Multivariate analysis of stoichiometric relations in the four taxa of spiders (PCA). ANOVA performed for the 1st axis scores. Different letters denote significant differences in multielemental stoichiometry between taxa (F = 18.70, *p* = 0.02). Bars denote the confidence intervals. Table 1 provides information on the contribution of different elements to the scores of Axis 1.

**Figure 4 insects-11-00484-f004:**
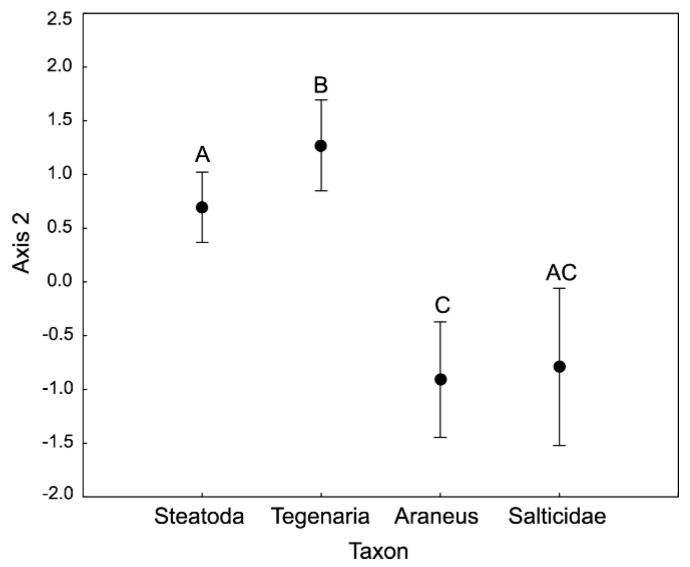
Multivariate analysis of stoichiometric relations in the four taxa of spiders (PCA). ANOVA performed for the 2nd axis scores. Different letters denote significant differences in multielemental stoichiometry between taxa (F = 3.78, *p* < 0.00001). Bars denote the confidence intervals. Table 1 provides information on the contribution of different elements to the scores of Axis 2.

**Table 1 insects-11-00484-t001:** Loadings values for all elements used in the PCA that variously contributed to the pattern observed in Figure 1. The mark ‘− ‘ denotes vectors’ directions.

	Axis 1	Axis 2
Ca	−0.08	−0.42
Cu	−0.91	−0.28
Fe	−0.69	0.42
Mg	−0.33	0.54
Mn	−0.68	−0.63
K	−0.84	0.31
Na	−0.03	0.95
Zn	−0.76	0.04
P	−0.27	−0.21
N	−0.93	−0.06
C	0.82	−0.05

**Table 2 insects-11-00484-t002:** Two-way ANOVA performed separately for 1st and 2nd axis scores. Statistically significant results are bolded.

1st Axis	Df	SS	MS	F	*p*
Taxon	**3**	**4.25**	**1.42**	**3.78**	**0.02**
Sex	**1**	**1.67**	**1.67**	**4.46**	**0.04**
Taxon × Sex	3	2.57	0.86	2.28	0.10
Error	34	12.75	0.37		
Total	41	42.00			
**2nd Axis**	**Df**	**SS**	**MS**	**F**	*p*
Taxon	**3**	**14.58**	**4.86**	**18.70**	**<0.00001**
Sex	1	0.51	0.51	1.98	0.17
Taxon × Sex	3	1.81	0.60	2.32	0.09
Error	34	8.84	0.26		
Total	41	42.00

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
