# Peer review of "Sexual Dimorphism in the Multielemental Stoichiometric Phenotypes and Stoichiometric Niches of Spiders"

_insects, 2020, doi:10.3390/insects11080484_

Round 1

Reviewer 1 Report

I thought this study was interesting and well written. The study design and analysis seemed strong. I had some concerns but all of these comments can be addressed through minor changes in the text.

My biggest criticism concerns the use of animal body contents to measure niches or nutrient requirements. For animals that are growing homeostatically, it is both the elemental phenotype and the turnover rates of nutrients that determine demand for resources. It is important to acknowledge that turnover rates vary for different nutrients and, hence, nutritional requirements are not proportional to current body contents. In the most basic example, animals require energy to digest and assimilate nutrients (in metabolic ecology, this is often referred to as Specific Dynamic Action). Energy preferentially comes from carbon-containing molecules. Hence, to create tissue with a certain amount of C, N and P, an animal needs those amounts of C, N and P plus an extra amount of C used to fuel the metabolic activities involved in creating those tissues. I believe that Sterner and Elser mention this in their theories, but many people that use the theories fail to recognize the importance of nutrient turnover rates. Because of this, comparison of an animal’s body and its food (e.g., stoichiometric mismatch) cannot be used as a measure of nutrient limitation. Also, we don’t know if turnover rates differ between the sexes. Males might have higher turnover of C because they are using it as fuel to locate females, while females may primarily use C as a nutrient invested in eggs or their own lipid reserves.

In addition, elemental phenotype (or current body contents) can only be used as a predictor of dietary requirements if the animal is building body tissues (the animal is growing). For reproductive animals, dietary requirements are determined by the nutrient content of the reproductive tissues and maintenance metabolism of the animal. Females need nutrients to build eggs and males often need nutrients to build sperm and for energy to locate females.

So, I think it’s great that you’re studying variation in nutrient content of different sexes but we are far off from using that information to say anything about nutrient limitation of adult males versus females because we cannot use current body contents as a surrogate for dietary requirements of adults (because it doesn’t take into account turnover or the nutrient requirements of reproductive tissues like eggs). I think it’s OK to mention the potential implications of the differences in elemental composition for nutrient limitation but I think the authors should add caveats (nutrient turnover and nutrient requirements of reproductive tissue) in the introduction and discussion.

More detailed comments:

Line 108: What were the reproductive states of the adult females? Were they full of eggs or had they recently laid eggs? It doesn’t seem like you would know because you would have to dissect the spider to tell. The reproductive state of the spider can significantly affect their elemental phenotype because 1) the elemental composition of eggs likely differs from that of the female body and 2) eggs can comprise a significant proportion of female body mass. If you don’t know, then at least clearly state these issues and that future studies should better study how reproductive state affects elemental content.

Line 139: Please state why you used a log transformation.

Line 146: Can you provide a loading table for axes 1 and 2? It would be easier to see and compare than having the loadings in the text.

Results: For figures 2-4, you discuss differences in individual elements in the text of the results (like the last sentence of the results) but only show figures for the comparisons of the principal component axes. The text should match the figures. If you didn’t statistically analyze individual elements, then you should only be discussing the axes and not the elements.

Author Response

We would like to express our great appreciation for the comments on our paper.

Reviewer 1:

My biggest criticism concerns the use of animal body contents to measure niches or nutrient requirements. For animals that are growing homeostatically, it is both the elemental phenotype and the turnover rates of nutrients that determine demand for resources. It is important to acknowledge that turnover rates vary for different nutrients and, hence, nutritional requirements are not proportional to current body contents. In the most basic example, animals require energy to digest and assimilate nutrients (in metabolic ecology, this is often referred to as Specific Dynamic Action). Energy preferentially comes from carbon-containing molecules. Hence, to create tissue with a certain amount of C, N and P, an animal needs those amounts of C, N and P plus an extra amount of C used to fuel the metabolic activities involved in creating those tissues. I believe that Sterner and Elser mention this in their theories, but many people that use the theories fail to recognize the importance of nutrient turnover rates. Because of this, comparison of an animal’s body and its food (e.g., stoichiometric mismatch) cannot be used as a measure of nutrient limitation. Also, we don’t know if turnover rates differ between the sexes. Males might have higher turnover of C because they are using it as fuel to locate females, while females may primarily use C as a nutrient invested in eggs or their own lipid reserves.

In addition, elemental phenotype (or current body contents) can only be used as a predictor of dietary requirements if the animal is building body tissues (the animal is growing). For reproductive animals, dietary requirements are determined by the nutrient content of the reproductive tissues and maintenance metabolism of the animal. Females need nutrients to build eggs and males often need nutrients to build sperm and for energy to locate females.

So, I think it’s great that you’re studying variation in nutrient content of different sexes but we are far off from using that information to say anything about nutrient limitation of adult males versus females because we cannot use current body contents as a surrogate for dietary requirements of adults (because it doesn’t take into account turnover or the nutrient requirements of reproductive tissues like eggs). I think it’s OK to mention the potential implications of the differences in elemental composition for nutrient limitation but I think the authors should add caveats (nutrient turnover and nutrient requirements of reproductive tissue) in the introduction and discussion.

Response:

We agree with the reviewer. Stimulated by the reviewer’s suggestion, we have added relevant information to the Introduction, Methods and Discussion (lines 51-63, 136-139, 227-233). We now acknowledge that a full understanding of the limitations by elements requires additional information about the nutrient turnover rates and reproductive activity requirements.  

Line 108: What were the reproductive states of the adult females? Were they full of eggs or had they recently laid eggs? It doesn’t seem like you would know because you would have to dissect the spider to tell. The reproductive state of the spider can significantly affect their elemental phenotype because 1) the elemental composition of eggs likely differs from that of the female body and 2) eggs can comprise a significant proportion of female body mass. If you don’t know, then at least clearly state these issues and that future studies should better study how reproductive state affects elemental content.

Response:

We agree. This information has now been added in Materials and Methods (lines 136-139), and we refer to this issue in the Discussion (lines 227-233).

Line 139: Please state why you used a log transformation.

Response:

This transformation was dictated by the need to normalize data with different ranges of absolute values (and units). At the same time, the transformation helped us satisfy the assumption of normality required by our statistical methods.

Line 146: Can you provide a loading table for axes 1 and 2? It would be easier to see and compare than having the loadings in the text.

Response:

We agree, and this information is now in Table 1.

Results: For figures 2-4, you discuss differences in individual elements in the text of the results (like the last sentence of the results) but only show figures for the comparisons of the principal component axes. The text should match the figures. If you didn’t statistically analyze individual elements, then you should only be discussing the axes and not the elements

Response:

We feel that there is no problem here because each axis “means something” in terms of the amount of each element. For simplicity, when discussing differences in the values of scores provided by the axes, we refer directly to the elements that mainly contributed to the axes (i.e., had the greatest impact on the values of the axis scores). Thus, our results are reported in a more straightforward and biologically relevant manner. Please note that specific elements with high loading values for a particular axis contribute to the observed differentiation and the variance explained by each axis. Therefore C, N, Zn, K and Cu (but not Na and Ca) may be discussed in relation to the scores of the 1st axis and Na, Ca and Mn may be discussed in relation to the scores of the 2nd axis.

Reviewer 2 Report

It was a pleasure to read this well-written and insightful work. The framework, aim, and hypotheses are well reasoned. The model organism (spiders) is particularly suited due to the often pronounced sexual dimorphism in most spider species. In this respect, I think it could worth mentioning this peculiarity when introducing the model system by referring to some recent references (e.g., Cordellier, M., Schneider, J. M., Uhl, G., & Posnien, N. (2020). Sex differences in spiders: from phenotype to genomics. Development Genes and Evolution, 1-18 ; McLean, C. J., Garwood, R. J., & Brassey, C. A. (2018). Sexual dimorphism in the Arachnid orders. PeerJ, 6, e5751).

Find below a few minor suggestions.

LINE COMMENTS:

-L95-96: “species” instead of “taxa”?

-L110-120: why most species were identified only at the genus level?

- Figure 1. It seems that most of the difference in the stoichiometric niche between sexes is driven by a single species, Steatoda grossa, having male and females far apart in the multidimensional space. I’m wondering why this is the case... it could be worth to discuss it.

-Table 1. “<0.00000” doesn’t make sense from a mathematical standpoint. I would write “<0.00001”, or round to the first few decimals “<0.001”

-L230: I think the hyphen between too and small is not needed.

-A final, extemporaneous comment. Characterizing the niche with a PCA and using ANOVA to test for differences is perfectly fine. That being said, in my opinion it would be particularly interesting to describe the stoichiometric niche using more sophisticate tools that allow to explicitly test for the overlap between niches (e.g., hypervolumes; https://besjournals.onlinelibrary.wiley.com/doi/full/10.1111/2041-210X.13424). I’m not expecting you to include this, it’s just a possibility to think about (I also don’t really know the literature on stoichiometric niche: perhaps this is routinely done already).

Author Response

We would like to express our great appreciation for the comments on our paper.

Reviewer 2:

… I think it could worth mentioning this peculiarity when introducing the model system by referring to some recent references (e.g., Cordellier, M., Schneider, J. M., Uhl, G., & Posnien, N. (2020). Sex differences in spiders: from phenotype to genomics. Development Genes and Evolution, 1-18 ; McLean, C. J., Garwood, R. J., & Brassey, C. A. (2018). Sexual dimorphism in the Arachnid orders. PeerJ, 6, e5751).

Response:

We agree. We now consider this topic in lines 99-100.

L95-96: “species” instead of “taxa”?

Response:

We were not able to identify all specimens to a species; therefore, certain specimens are grouped according to a higher taxon (genus). Consequently, we use the term “taxon” when we refer to our groups.

Figure 1. It seems that most of the difference in the stoichiometric niche between sexes is driven by a single species, Steatoda grossa, having male and females far apart in the multidimensional space. I’m wondering why this is the case... it could be worth to discuss it.

Response:

Similar differentiation is visible in the case of Tegenaria. We added a brief discussion of this issue (lines 227-233); however, a thorough discussion of this issue was impossible (due to the scarce data).  

Table 1. “<0.00000” doesn’t make sense from a mathematical standpoint. I would write “<0.00001”, or round to the first few decimals “<0.001”

Response:

We agree. This value is now corrected to <0.00001.

L230: I think the hyphen between too and small is not needed.

Response:

We have removed the hyphen.

A final, extemporaneous comment. Characterizing the niche with a PCA and using ANOVA to test for differences is perfectly fine. That being said, in my opinion it would be particularly interesting to describe the stoichiometric niche using more sophisticate tools that allow to explicitly test for the overlap between niches (e.g., hypervolumes; https://besjournals.onlinelibrary.wiley.com/doi/full/10.1111/2041-210X.13424). I’m not expecting you to include this, it’s just a possibility to think about (I also don’t really know the literature on stoichiometric niche: perhaps this is routinely done already).

Response:

Thank you for this remark. We agree and will consider this approach in future studies. However, because our data in the current study have a relatively simple structure, we would like to retain our statistical approach, which will help to keep this paper within the framework of a simple and short Communication paper.